Analysis of microRNA expression profiles in exosomes derived from acute myeloid leukemia by p62 knockdown and effect on angiogenesis

Li Chuan 1
Long Xinyi 1
Liang Peiqi 2
Liu Zhuogang 1
Wang Chen 1
Hu Rong hur@sj-hospital.org 1
1 Hematology Department, Shengjing Hospital of China Medical University , Shenyang , China
2 Hematology Department, The First Affiliated Hospital of Suzhou University , Suzhou , China
Uversky Vladimir
Electronic publication date: 2022 Jul 22
Publication date: 2022
Volume: 10
Electronic Location ID: e13498
Received 2022 Jan 3; Accepted 2022 May 5
Copyright: ©2022 Li et al.
Copyright year: 2022
Copyright holder: Li et al.
License: This is an open access article distributed under the terms of the Creative Commons Attribution License, which permits unrestricted use, distribution, reproduction and adaptation in any medium and for any purpose provided that it is properly attributed. For attribution, the original author(s), title, publication source (PeerJ) and either DOI or URL of the article must be cited.
License URL: https://creativecommons.org/licenses/by/4.0/

Keywords: Exosome, microRNA, Acute myeloid leukemia, P62, Angiogenesis

Funding: Scientific research project of the Education Department of Liaoning Province (2019 64) This project is supported by the Scientific research project of the Education Department of Liaoning Province (2019 64). The funders had no role in study design, data collection and analysis, decision to publish, or preparation of the manuscript.

==============================
Objectives

In this study, we aimed to investigate the effect of p62 on angiogenesis and microRNA (miRNA) expression profiles in acute myeloid leukemia (AML) exosomes.

Methods

An Exiqon v19.0 microRNA MicroArray was used to profile miRNAs in exosomes derived from parental and p62-knockdown U937 cells. The Gene Ontology (GO) and Kyoto Encyclopedia of Genes and Genomes (KEGG) databases were used to predict the biological functions and potential mechanisms of differentially expressed miRNAs in AML exosomes. Endothelial cell tube formation assays using human umbilical vein endothelial cells (HUVECs) were performed to investigate the effect of AML exosomes on angiogenesis.

Results

We demonstrated that 2,080 miRNAs were expressed in exosomes derived from our cultured cell samples, of which 215 and 208 miRNAs were upregulated and downregulated, respectively, in p62-knockdown U937 cells (fold change ≥ 2, P < 0.05). GO analysis indicated that miRNAs were most enriched in the intercellular pathways. Biological process analysis revealed that 1460 biological processes were associated with downregulated transcripts, including 19 pathways related to vesicles, and 1,515 pathways were upregulated, including 8 pathways related to vesicles. Molecular function analysis indicated that protein binding, transcription regulator activity, and DNA-binding transcription factor activity were enriched (P < 0.05). Pathway analysis indicated that 84 pathways corresponded to upregulated transcripts, and 55 pathways corresponded to downregulated transcripts (P < 0.05). We also found that exosomes derived from U937 cells promoted angiogenesis in HUVECs.

Conclusions

Our data suggest that exosomal miRNAs may play important roles in the pathogenesis of AML, which may be treated by p62 knockdown with exosomal miRNAs to inhibit angiogenesis.

Introduction

Acute myeloid leukemia (AML) is a fatal hematological malignancy with high recurrence rate. For patients receiving the most intensive treatment, the overall 5-year survival rate remains below 50%. For the remaining patients, the prognosis is even worse (Burnett, Wetzler & Lowenberg, 2011). AML is characterized by multiple recurring mutations. These mutations affect disease response to treatment and the risk of recurrence (Patel et al., 2012; Short, Rytting & Cortes, 2018; Staudt et al., 2018; Xie et al., 2014). P62 regulates cell survival and death via various signal transduction pathways (Komatsu, Kageyama & Ichimura, 2012). A study found that upregulated p62 expression could promote AML cell maturation into granulocytes, depending on NF-κB activation, predicting poor AML prognosis (Trocoli et al., 2014). In addition, loss of p62 impaired leukemia cell growth and colony formation and prolonged the development of leukemia in mice (Nguyen et al., 2019). High expression of the selective autophagy receptor p62 is associated with a poor prognosis in AML (Nguyen et al., 2019).

Exosomes are nanometer-scale extracellular vesicles containing many microRNAs (miRNAs) that are secreted from cells in both normal and pathological conditions (Skog et al., 2008). It has been found that normal hematopoietic stem cell proliferation and differentiation are suppressed by exosomes releasing miR-150 and miR-155 through c-MYB inhibition. In this manner, a malignant phenotype constitutes perpetual existence by changing hematopoietic stem cell biological behaviors (Hornick et al., 2016). This finding suggests a biological role for such miRNAs in malignant tumor progression (De Kouchkovsky & Abdul-Hay, 2016). Leukemic cells can stimulate neovascularization in the bone marrow (Wang et al., 2019) and secrete angiogenic factors, indicating adverse prognosis in AML (Haouas, 2014). Exosomes may accelerate angiogenesis and promote tumor progression by harboring miRNAs (Aslan et al., 2019). Thus, exosomal miRNAs may be new targets for AML treatment. However, regulation of angiogenesis by p62 remains largely unknown.

In this study, we constructed a miRCURYTM LNA Array (v.19.0) of miRNAs in exosomes derived from AML cells after p62 knockdown. The miRNAs in exosomes were analyzed by identifying signature miRNAs. We then investigated angiogenesis in human umbilical vein endothelial cells (HUVECs) exposed to exosomes derived from parental U937 cells, p62-knockdown U937 cells, or control cells. The data from these studies may shed light on the relationship between exosomal miRNAs and AML, further enhancing our understanding of AML progression.Our study may aid the development of potential biomarkers for the diagnosis and prognosis of AML progression.

Materials and Methods

Cell culture and lentiviral vector cell line construction

The human acute monocytic leukemia cell line U937 was purchased from the Bena Culture Collection (Beijing, China) and stored in our laboratory. A recombinant lentivirus vector-mediated SQSTM1 gene (LV-SQSTM1-RNAi) and an empty recombinant adenovirus vector (Hu6-MCS-CMV-EGFP) were constructed. U937 cells were placed in a six-well plate. Polybrene (Gene, Shanghai, China) was used for the transfection. The transfection system included 1.8 mL RPMI-1640 with 10% fetal bovine serum, 10 µL LV-SQSTM1-RNAi or Hu6-MCS-CMV-EGFP, and 0.9 µL polybrene. After transfection for 24 h, the cell suspension was collected and centrifuged at 800 rpm for 5 min, the supernatant was discarded, and two mL of RPMI-1640 with 10% fetal bovine serum was added. After transfection for 48 h, fluorescence was observed. After culturing, 5 µg/mL puromycin was added, and the cells were screened for 15 days. After selecting the surviving cells, cell lines with clonal stability were cryopreserved and characterized through RT–qPCR and western blotting.

Cell viability cell counting kit (CCK)-8

U937 cells were plated at a density of 3–5 × 104 cells/well in 96-well plates and allowed to grow for 12, 24, and 48 h. Next, 10 µL CCK-8 solution (Promega, Madison, WI, USA) was added to the cell suspension and incubated for 2 h. Absorbance was measured using a spectrophotometer at 450 nm. The experiment was repeated at least three times.

Flow cytometric cell apoptosis

Flow cytometry analysis using an Annexin V-FITC/PI detection kit was used to compare the apoptosis rate of p62-control and p62 knockdown U937 cells. After 48 h of incubation, the cells were washed with phosphate-buffered saline and resuspended in 400 µL of 1 × binding buffer. Thereafter, 5 µL Annexin V-FITC and 5 µL PI were added to the mixture and stained in the dark for 15 min at room temperature. Apoptosis was detected using flow cytometry (Beckman Coulter, La Brea, CA, USA) immediately after staining.

Exosome collection and identification

P62-siRNA coated with lentivirus interfered with U937 cells to downregulate the expression of p62, with the empty virus vector used as a control. Two groups of cells were used as follows: p62-knockdown U937 cells and controls. The two groups of cells were cultured for 48 h in serum-free media. The supernatant was collected for exosome extraction via ultracentrifugation. Exosome shape and size were observed using electron microscopy, and exosomal markers were detected using western blot analysis.

Western blot analysis

Total protein was extracted from U937 cells that had or had not been transfected with the p62-encoding gene using radioimmunoprecipitation assay lysis buffer (Beyotime, Shanghai, China). The protein concentration was determined using the BCA method. Equal amounts of protein samples were added to each well, separated using 10% SDS-PAGE, and transferred to a polyvinylidene chloride transfer membrane (Merck Millipore, Burlington, MA, USA). The membrane was blocked with 5% skimmed milk for 2 h. It was then washed with TBST and incubated with primary antibodies against p62, TSG101, CD63, CD9, calnexin, and GAPDH (all from Abcam, Cambridge, UK; 1:1000) overnight at 4 °C. Thereafter, the membrane was incubated with anti-rabbit or anti-mouse horseradish peroxidase-conjugated secondary antibodies at room temperature for 2 h after washing three times with TBST. An enhanced chemiluminescence substrate (Thermo Fisher Scientific) was used to detect the protein bands. Image Lab software was used to detect and analyze the density of each band (Bio-Rad, Hercules, CA, USA).

RNA extraction, miRNA labeling and array hybridization

TRIzol (Invitrogen, Carlsbad, CA, USA) was used to extract total RNA. A NanoDrop spectrophotometer (ND-1000; NanoDrop Technologies, Wilmington, DE, USA) was used to measure RNA quality and quantity. RNA integrity was assessed using gel electrophoresis. After quality control, miRNA labeling was performed according to the instructions of the miRCURY™ Hy3™/Hy5™ Power Labeling Kit (Exiqon, Vedbaek, Denmark). First, 1 µL RNA in 2 µL water was mixed with 1 µL CIP buffer and CIP (Exiqon, Vedbaek, Denmark). The mixture was then incubated at 37 °C for 30 min. The mixture was incubated at 95 °C for 5 min to stop the reaction. Then, 3 µL labeling buffer, 1.5 µL fluorescent label (Hy3TM), 2 µL dimethyl sulfoxide, and 2 µL labeling enzyme were added. The mixture was then incubated for 1 h at 16 °C, followed by 15 min at 65 °C to terminate the reaction. Hy3-labeled samples were hybridized on the miRCURYTM LNA array (v.19.0; Exiqon, Vedbaek, Denmark) according to the manufacturer’s instructions. A total of 25 µL Hy3™-labeled samples and 25 µL hybridization buffer were denatured at 95 °C for 2 min and then incubated on ice for 2 min. The hybridization system (Nimblegen Systems, Inc., Madison, WI, USA) was used with the microarray set at 56 °C for 16 to 20 h. After hybridization, the slides were washed several times using a washing buffer kit (Exiqon, Vedbaek, Denmark). Finally, an Axon GenePix 4000 B microarray scanner (Axon Instruments, Foster City, CA, USA) was used to scan the slides.

miRNA array scanning and analysis

GenePix Pro 6.0 software (Axon, Instruments, Foster City, CA, USA) was used to extract data by analyzing the imported scanned images. The samples were chosen to calculate normalization factors if the replicated miRNAs were averaged and for miRNAs with intensities ≥30. Median normalization was used to normalize the data. Normalized data = (foreground background)/median; the median was the 50% quantile of miRNA intensity, which was larger than 30 in all samples after background correction. After normalization, the miRNAs with significant differences between the two groups were determined according to the fold change and P value. Finally, hierarchical clustering was used to show the different miRNA expression profiles between the samples.

MiRNA and mRNA networks and prediction of miRNA function

TargetScan7.1 (http://www.targetscan.org/) and mirdbV5 (http://mirdb.org/) are online sites for miRNA target gene prediction (Agarwal et al., 2015; Chen & Wang, 2020). In our research, we used two databases to predict the target genes of differentially expressed miRNAs: targetscan7.1 and mirdbV5.

Gene ontology (GO) and kyoto encyclopedia of genes and genomes (KEGG) pathway analyses of differentially expressed miRNAs

We used the GO (http://www.geneontology.org) and KEGG (http://www.genome.ad.jp/kegg/) databases to study the potential organisms and signaling pathways of differentially expressed miRNAs. Differences were considered statistically significant at P < 0.05.

Endothelial cell tube formation assays

HUVECs were exposed to exosomes derived from parental U937 cells, p62-knockdown U937 cells, or p62-control cells when cultured in 1640 medium. Cells were plated in 96-well plates with 50 µL Matrigel (BD Biosciences), seeded at 3 × 104 cells per well. Tubules were photographed using phase microscopy after incubation for 0, 3, and 6 h at 37 °C with 5% CO2.

Statistical analysis

We used GraphPad Prism 6 to corroborate the statistical significance of the data for all graphs in this study. Values are presented as the means ± standard deviation. Differences between groups were analyzed using Student’s t-test, and statistical significance was set at P < 0.05.

Results

Confirmation of knockdown p62 in U937 cell line

To generate p62-knockdown U937 cells, we used a lentivirus to transfect cells and observed the transfection efficiency by fluorescence microscopy. We then conducted RT–qPCR analysis to determine the expression levels of the autophagy gene encoding p62 (Figs. 1A–1B) and western blotting to detect the expression levels of the autophagy core protein p62 (Figs. 1C–1D). Notably, we found that the p62 gene and protein expression levels in U937 cells were successfully reduced compared with controls after LV-SQSTM1-RNAi treatment. CCK8 (Fig. 1E), and flow cytometry (Figs. 1F–1G), show that p62 knockdown inhibits the proliferation of U-937 cells and promotes apoptosis.

Figure 1 The expression levels of thep62 gene and protein in U937 cells were successfully reduced compared with controls after LV-SQSTM1-RNAi treatment.

(A) Transfection efficiency of U937 cells by fluorescence microscopy. (B) Relative mRNA expression of p62 by reverse transcription (RT)–quantitative (q)PCR. (C–D) Protein expression of p62 by western blotting. (E) Cell proliferation measured by cell counting kit (CCK)-8. (F) Apoptotic rates of cells. (G) Typical flow cytometry dot plot diagrams of cells. U937, parental cells; p62-con, U937 cells infected by an empty recombinant adenovirus vector; p62-, U937 cells with p62 knockdown. N = 3. Data were shown as means ± SD. *P < 0.05.

Characteristics of exosomes

We observed exosome formation using electron microscopy after ultracentrifugation. Electron microscopy revealed exosomes as vesicles with a double-layer membrane structure (30–100 nm in diameter, Fig. 2A). Western blotting indicated that exosomes expressed TSG101, CD63, and CD9, but not calnexin (Fig. 2B). Exosome concentrations and quantities were measured using BCA. The concentrations of parental, control, and p62-knockdown U937 cells were 2.248 ± 0.245, 2.212 ± 0.092, and 2.189 ± 0.102, respectively (Figs. 2C, 2D). We found that the quantity of exosomes in p62-knockdown U937 cells was lower than that in the controls (P < 0.05).

Figure 2 The exosome forms by electron microscopy after ultracentrifugation.

(A) Exosomes from U937 cells with p62 knockdown and parental cells by scanning electron microscope. (B) Western blot results of exosomal protein expression from three cell lines. (C) Concentrations of exosomes in three cell lines. (D) Quantities of exosomes in three cell lines. U937, parental cells; p62-con, U937 cells infected by an empty recombinant adenovirus vector; p62-, U937 cells with p62 knockdown. N = 3. Data were shown as means ± SD. **P < 0.01.

Differentially expressed miRNAs in exosomes and PCR verification

We chose differentially expressed miRNAs in exosomes based on the fold change in the two samples. Microarray analysis revealed that 2,080 miRNAs were expressed in exosomes, 215 were upregulated, and 208 were downregulated (fold change ≥ 2, P < 0.05, Fig. 3A) in p62-knockdown and control U937 cells. We found that miR-3064-3p, one of the top expressed miRNAs, and miR-339-5p, which was the most uniformly expressed endogenous control reference gene, were downregulated in U937 cells with p62 knockdown. RT–qPCR was used to detect the expression of exosomal miRNAs in the cells, revealing that the expression levels of miR-3064-3p and miR-339-5p decreased in U937 cells with p62 knockdown, consistent with the microarray profile (Fig. 3B).

Figure 3 The differentially expressed miRNAs in exosomes.

(A) A heat map of U937 cells (p62 knock-down and control). The red parts indicate a high expression level and blue corresponds to a low expression level. (B) Relative expression of miRNA-3064-3p and miR-339-5p. U937, parental cells; p62-con, U937 cells infected by an empty recombinant adenovirus vector; p62-, U937 cells with p62 knockdown. N = 3. Data were shown as means ± SD. *P < 0.05.

Figure 4 Gene ontology (GO) analysis of the top 50 miRNAs for upregulation and downregulation in U937 cells with p62 knockdown and controls, respectively.

(A) The most highly enriched GO terms for differentially expressed transcripts for upregulation. (B) The most highly enriched GO terms for differentially expressed transcripts for downregulation. The most highly enriched GO terms for upregulated transcripts: (C) biological process (BP); (E) cellular component (CC); and (G) molecular function (MF). The most highly enriched GO terms for downregulated transcripts: (D) biological process (BP); (F) cellular component (CC); and (H) molecular function (MF).

Figure 5 Corresponding down-regulated pathways and network of miRNAs and mRNAs.

(A) Corresponding pathways of downregulation; (B) Venn diagram of miRNA co-expression in downregulation between targetscan7.1 and mirdbV5; (C) network of miRNAs and mRNAs in downregulation.

Figure 6 Corresponding up-regulated pathways and network of miRNAs and mRNAs.

(A) Corresponding pathways of upregulation; (B) Venn diagram of miRNA co-expression in upregulation between targetscan7.1 and mirdbV5; (C) network of miRNAs and mRNAs in upregulation.

GO and KEGG pathway analyses of differentially expressed miRNAs

We used GO and KEGG pathways to analyze the top 50 miRNAs that were upregulated and downregulated in p62-knockdown and control U937 cells, respectively. We focused on cellular component (CC) (Figs. 4E–4F), biological process (BP) (Figs. 4C–4D), and molecular function (MF) (Figs. 4G–4H) targeted by upregulated and downregulated genes. We found that the neuron part was the top part of gene expression in downregulated genes, including 233 genes (Fig. 4B), and that the intercellular part was the top part of upregulated genes, including 1,843 genes (Figs. 4G, 4E). According to the BP analysis, 1,460 BPs were associated with downregulated transcripts, 19 pathways were related to vesicles, 1,515 pathways were related to upregulated transcripts, and 8 pathways were related to vesicles. MF analysis revealed that 1,544 upregulated genes and 1,200 downregulated genes were involved in protein binding (Figs. 4G–4H). KEGG pathway analysis indicated that 84 pathways corresponded to upregulated transcripts, and 55 pathways corresponded to downregulated transcripts (P < 0.05) (Figs. 5A and 6A). For downregulated transcripts, the “TNF signaling pathway” (Pathway ID: hsa04668) that included 23 genes was most affected, followed by the “MAPK pathway” (Pathway ID: hsa04010) (Fig. 5A). As for upregulated transcripts, the “PI3K–Akt signaling pathway” (Pathway ID: hsa04151) that included 80 upregulated genes was the most enriched pathway (Fig. 6A). The miRNAs of exosomes derived from U937 cells (p62-knockdown and controls) influenced gene expression and signaling pathways that participated in vesicle formation and AML generation (Figs. 5A and 6A).

MiRNA and mRNA networks and prediction of miRNA function

TargetScan7.1 (http://www.targetscan.org/) and mirdbV5 (http://mirdb.org/) were employed to predict the potential target genes of miRNAs. We found that 2,018 genes co-expressed with upregulated miRNAs and 2,749 genes co-expressed with downregulated miRNAs in the two databases (Figs. 5B and 6B). Besides, networks of miRNAs and mRNAs in upregulation and downregulation were shown in the Figs. 5C and 6C.

HUVEC angiogenesis

HUVECs were inoculated in Matrigel and incubated with AML exosomes for a certain period of time to determine whether exosomes from AML cells could induce HUVEC tubular differentiation in vitro. A CCK-8 assay was used to detect cell proliferation after 12, 24 and 48 h. The proliferation of cell lines was not statistically significant (Fig. 7A). Microscopic analysis showed that exosomes from U937 cells induced the formation of a tubular network, compared with p62-knockdown and p62-control cells (Figs. 7B–7C). By counting the node number, junction number and segment length, we found that the HUVECs with U-937 cells and exosomes generated more blood vessels (p < 0.05). Among the three groups of HUVECs+Exo(U937), HUVECs+Exo(p62-con) and HUVECs+Exo(p62-), the HUVECs+Exo(p62-) group grew slower than the other two groups (p < 0.05). In summary, the above results indicated that the fastest angiogenesis occurred in the presence of exosomes from U937 cells; the slowest angiogenesis occurred in the presence of exosomes from p62-knockdown U937 cells.

Figure 7 HUVECs were co-cultured with exosomes secreted by AML cells.

(A) A cell counting kit (CCK)-8 assay was used to detect cell proliferation after 12, 24 and 48 h. The proliferation of cell lines was not statistically significant; (B) 0, 3 and 6 h to detect HUVEC angiogenesis. We found the exosomes of U937 cells can promote the angiogenesis of HUVECs. (C) Quantitative results of 6 h angiogenesis assay. EXO, exosomes; HUVECs, human umbilical vein endothelial cells; AML, acute myeloid leukemia; U937, parental cells; p62-con, U937 cells infected by an empty recombinant adenovirus vector; p62-, U937 cells with p62 knockdown. N = 3. Data were shown as means ± SD. *P < 0.05.

Discussion

Although advances have been made in AML supportive care, prognostic risk stratification, and established therapies, patients with AML have poor long-term prognosis (De Kouchkovsky & Abdul-Hay, 2016). The identification of several gene mutations can guide treatment, such as PML-RARA for acute promyelocytic leukemia (APL). P62 is an autophagy receptor and a selective adaptor protein. In addition, p62 is involved in many signal transduction pathways, including the Keap1–Nrf2 pathway that plays a critical role in proteasomal degradation of ubiquitinated proteins and autophagy progression (Liu et al., 2016). Ser407 of the ubiquitin-associated (UBA) domain of p62 is phosphorylated by autophagy-associated protein 1 (ATG1/ULK1), followed by phosphorylation at Ser403 of the UBA domain by casein kinase 2 or TANK-binding kinase 1 that leads to p62 ubiquitination and promotes autophagic degradation (Myeku & Figueiredo-Pereira, 2011). The PB1 domain of p62 promotes packaging of ubiquitinated substrates by self-oligomerization and transports packaged substrates to the autophagy pathway to participate in autophagosome formation (Ichimura et al., 2008). P62 interacts with LC3, an autophagosome marker protein, to form a complex through its LIR domain that is degraded in autophagosomes as an autophagy-specific substrate (Pankiv et al., 2007). A study has shown that mutations in SQSTM1, which codes for p62, are a causative factor in Paget disease of bone, as well as amyotrophic lateral sclerosis and frontotemporal dementia (Le Ber et al., 2013). Abnormal amplification and phosphorylation of p62/SQSTM1 have been implicated in tumor progression and resistance therapy, such as in hepatocellular carcinoma (Saito et al., 2016), and platinum-resistant cells of high-grade serous ovarian cancer (Nguyen et al., 2017). P62/SQSTM1 upregulation promotes granulocytic differentiation and survival of AML cells through a mechanism that depends on NF-κB activation (Trocoli et al., 2014).

Accumulating evidence indicates that exosomes in tumors are oncogenic. Crosstalk between bone marrow tumors and endothelial cells can affect tumor progression in hematological tumors, and exosomes containing miRNAs are crucial in bone marrow angiogenesis promotion in hematological tumors. By delivering miR-365, exosomes can mediate the horizontal transfer of drug resistance in chronic myeloid leukemia cells (Saito et al., 2016). K562 cells secrete exosomes containing miR-92a, which can enhance angiogenesis; miR-135b, which targets inhibitory hypoxia-inducible factor 1 angiogenesis, also enhances exocytosis secretion by multiple myeloma cells (Ohyashiki, Umezu & Ohyashiki, 2016).

In our study, p62 knockdown in U937 cells inhibited cell proliferation and promoted apoptosis (P < 0.05). We used microarray technology to study the expression patterns of exosomal miRNAs derived from two U937 cell lines to further explore the relationship between exosomal miRNAs and p62. We identified 2,080 miRNAs, including 215 upregulated and 208 downregulated miRNAs in p62-knockdown U937 cells. To further validate microarray analysis results, we performed RT–qPCR to validate the downregulated expression of miR-3064-3p and miR-339-5p in the same series of samples. Our future studies will involve identification of other miRNAs whose expression was the highest in our microarray results. The RT–qPCR results were consistent with those obtained from microarray analysis. In addition, KEGG pathway analysis revealed that 84 pathways corresponded to upregulated transcripts, and 55 pathways corresponded to downregulated transcripts. For the downregulated transcripts, the most affected pathway was the “TNF signaling pathway” (Pathway ID: hsa04668), followed by the “MAPK pathway” (Pathway ID: hsa04010). As for upregulated transcripts, the “PI3K–Akt signaling pathway” (Pathway ID: hsa04151) was the most enriched pathway.

We constructed a miRNA–mRNA correlation network that displayed 2,749 upregulated and 2,018 downregulated co-expressed genes in two databases. According to previous research, vascular endothelial growth factor A (VEGFA), VEGFC, GATA-binding protein 4 (Jia et al., 2018), matrix metallopeptidase 2 (Sun et al., 2010), and zinc finger protein (Li et al., 2016), were involved in cell angiogenesis in the co-expression network. These results negatively correlated with those of hsa-miR-3064-3p and hsa-miR-339-5p in our profiles. Furthermore, hsa-miR-3622a-5p and hsa-miR-3064-3p displayed significant negative correlations with the Grb2-associated regulators of mitogen-activated protein kinase (MAPK) 1 and MAPK6, which participate in the MAPK signaling pathway. These findings indicate that hsa-miR-3064-3p, hsa-miR-339-5p, hsa-miR-3622a-5p, and their co-expressed coding genes in the MAPK signaling pathway may play a significant role in the angiogenesis of AML cells.

Although AML cells secrete angiogenic factors to remodel the vascular system and gain chemoresistance, anti-angiogenic drugs are generally ineffective in AML treatment. Wang et al. (2019) found that exosomes secreted by AML cells can enhance glycolysis-mediated vascular remodeling and chemoresistance. According to previous reports, wogonoside inhibits angiogenesis in solid tumors by blocking the JAK2-STAT3 pathway and inhibiting the development of hematologic malignancies in AML (Lin et al., 2019). To further validate the exosome and angiogenesis results in AML, we grew HUVECs in the presence of exosomes derived from parental U937 cells, p62-knockdown U937 cells, and p62-control cells. We found that angiogenesis was fastest in the presence of U937 exosomes and slowest in the presence of p62-knockdown U937 exosomes. This finding illustrates that exosomes derived from AML cells play an important role in the angiogenesis of HUVECs.

Conclusions

In conclusion, after a detailed examination of miRNA expression in exosomes derived from AML cells, we found that hsa-miR-3064-3p and hsa-miR-339-5p displayed downregulated expression in p62-knockdown cells, compared with control cells. In addition, we demonstrated that several differentially expressed exosomal miRNAs were closely related to multiple GO items and pathways involved in carcinogenesis, indicating that exosomal miRNAs play a key role in AML pathogenesis. This information may aid the development of potential biomarkers for diagnosis and prognosis of AML progression. In the present study, we also found that exosomes derived from AML cells promoted angiogenesis. However, the relationship between exosomal miRNAs and angiogenesis needs to be further investigated. These findings support the notion that promising new treatment strategies may be developed against AML, based on exosomal miRNA analysis.

Supplemental Information

Supplemental Information 1 Raw data exported from the P62 transfection

Click here for additional data file.

Supplemental Information 2 Raw data exported from the lentiviral transfection

Click here for additional data file.

Supplemental Information 3 Raw data exported from the western blot in our experiment

Click here for additional data file.

Supplemental Information 4 Raw data exported from the exosome extraction

Click here for additional data file.

Supplemental Information 5 Raw data exported from the flow cytometry in apoptosis assay

Click here for additional data file.

Supplemental Information 6 Raw data exported from the sequencing analysis of miRNA

Click here for additional data file.

Supplemental Information 7 Raw data exported from the CCK-8 assay

Click here for additional data file.

Supplemental Information 8 Raw data exported from endothelial cell tube formation assays at 0h

Click here for additional data file.

Supplemental Information 9 Raw data exported from PCR and CCK-8 after P62 knockdown

Click here for additional data file.

Supplemental Information 10 The first experiment results of raw data exported from endothelial cell tube formation assays in HUVEC at 3h

Click here for additional data file.

Supplemental Information 11 The first experiment results of raw data exported from endothelial cell tube formation assays in control group at 3h

Click here for additional data file.

Supplemental Information 12 Raw data exported from endothelial cell tube formation assays in HUVEC at 3 h (the first experiment pictures)

Click here for additional data file.

Supplemental Information 13 Raw data exported from endothelial cell tube formation assays in control group at 3 h (the first experiment pictures)

Click here for additional data file.

Supplemental Information 14 raw data

Click here for additional data file.

Supplemental Information 15 The second experiment results of raw data exported from endothelial cell tube formation assays in control group at 3h

Click here for additional data file.

Supplemental Information 16 Raw data exported from endothelial cell tube formation assays in HUVEC at 3 h (the third experiment pictures)

Click here for additional data file.

Supplemental Information 17 Raw data exported from endothelial cell tube formation assays in HUVEC at 3 h (the second experiment pictures, repeat the first time)

Click here for additional data file.

Supplemental Information 18 Raw data exported from endothelial cell tube formation assays in HUVEC at 3 h (the first experiment pictures, repeat the second time)

Click here for additional data file.

Supplemental Information 19 The first experiment results of raw data exported from endothelial cell tube formation assays in p62 knockdown group at 3h

Click here for additional data file.

Supplemental Information 20 The fourth experiment results of raw data exported from endothelial cell tube formation assays in p62 knockdown group at 3h

Click here for additional data file.

Supplemental Information 21 Raw data exported from endothelial cell tube formation assays in control group at 3 h (the second experiment pictures, repeat the first time)

Click here for additional data file.

Supplemental Information 22 Raw data exported from endothelial cell tube formation assays in control group at 3 h (the third experiment pictures, repeat the first time)

Click here for additional data file.

Supplemental Information 23 The second experiment results of raw data exported from endothelial cell tube formation assays in p62 knockdown group at 3 h (repeat the first time)

Click here for additional data file.

Supplemental Information 24 Raw data exported from endothelial cell tube formation assays in control group at 3 h (the second experiment pictures)

Click here for additional data file.

Supplemental Information 25 Raw data exported from endothelial cell tube formation assays in U-937 at 3 h (the first experiment pictures, repeat the second time)

Click here for additional data file.

Supplemental Information 26 Raw data exported from endothelial cell tube formation assays in control group at 3 h (the third experiment pictures, repeat the first time)

Click here for additional data file.

Supplemental Information 27 Raw data exported from endothelial cell tube formation assays in p62 knockdown group at 3 h (the third experiment pictures, repeat the first time)

Click here for additional data file.

Supplemental Information 28 Raw data exported from endothelial cell tube formation assays in p62 knockdown group at 3 h (the first experiment pictures, repeat the first time)

Click here for additional data file.

Supplemental Information 29 Raw data exported from endothelial cell tube formation assays in p62 knockdown group at 3 h (the fourth experiment pictures, repeat the first time)

Click here for additional data file.

Supplemental Information 30 Raw data exported from endothelial cell tube formation assays in p62 knockdown group at 3 h (the second experiment pictures, repeat the second time)

Click here for additional data file.

Supplemental Information 31 Raw data exported from endothelial cell tube formation assays in p62 knockdown group at 3 h (the second experiment pictures, repeat the first time)

Click here for additional data file.

Supplemental Information 32 Raw data exported from endothelial cell tube formation assays in p62 knockdown group at 3 h (the third experiment pictures, repeat the second time)

Click here for additional data file.

Supplemental Information 33 Raw data exported from endothelial cell tube formation assays in U-937 at 3 h (the first experiment data, repeat second time)

Click here for additional data file.

Supplemental Information 34 Raw data exported from endothelial cell tube formation assays in U-937 at 3 h (the third experiment data, repeat the first time)

Click here for additional data file.

Supplemental Information 35 Raw data exported from endothelial cell tube formation assays in control group at 6 h (the first experiment data, repeat the first time)

Click here for additional data file.

Supplemental Information 36 Raw data exported from endothelial cell tube formation assays in U-937 at 3 h (the second experiment pictures, repeat the first time)

Click here for additional data file.

Supplemental Information 37 Raw data exported from endothelial cell tube formation assays in U-937 at 3 h (the first experiment pictures, repeat the second time)

Click here for additional data file.

Supplemental Information 38 Raw data exported from endothelial cell tube formation assays in HUVEC at 6 h (the first experiment pictures, repeat the first time)

Click here for additional data file.

Supplemental Information 39 Raw data exported from endothelial cell tube formation assays in HUVEC at 6 h (the second experiment pictures, repeat the fifth time)

Click here for additional data file.

Supplemental Information 40 Raw data exported from endothelial cell tube formation assays in HUVEC at 6 h (the third experiment pictures, repeat the first time)

Click here for additional data file.

Supplemental Information 41 Raw data exported from endothelial cell tube formation assays from U-937 exosomes at 6 h (the first experiment data, repeat the first time)

Click here for additional data file.

Supplemental Information 42 Raw data exported from endothelial cell tube formation assays from U-937 exosomes at 6 h (the third experiment data, repeat the second time)

Click here for additional data file.

Supplemental Information 43 Raw data exported from endothelial cell tube formation assays in HUVEC at 6 h (the second experiment pictures, repeat the second time)

Click here for additional data file.

Supplemental Information 44 Raw data exported from endothelial cell tube formation assays in HUVEC at 6 h (the second experiment pictures, repeat the third time)

Click here for additional data file.

Supplemental Information 45 Raw data exported from endothelial cell tube formation assays in HUVEC at 6 h (the second experiment pictures, repeat the first time)

Click here for additional data file.

Supplemental Information 46 Raw data exported from endothelial cell tube formation assays in HUVEC at 6 h (the third experiment pictures, repeat the third time)

Click here for additional data file.

Supplemental Information 47 Raw data exported from endothelial cell tube formation assays from control group exosomes at 6 h (the first experiment pictures, repeat the first time)

Click here for additional data file.

Supplemental Information 48 Raw data exported from endothelial cell tube formation assays from control group exosomes at 6 h (the first experiment pictures, repeat the third time)

Click here for additional data file.

Supplemental Information 49 Raw data exported from endothelial cell tube formation assays from control group exosomes at 6 h (the first experiment pictures, repeat the second time)

Click here for additional data file.

Supplemental Information 50 Raw data exported from endothelial cell tube formation assays from control group exosomes at 6 h (the third experiment pictures, repeat the second time)

Click here for additional data file.

Supplemental Information 51 Raw data exported from endothelial cell tube formation assays from control group exosomes at 6 h (the third experiment pictures, repeat the first time)

Click here for additional data file.

Supplemental Information 52 Raw data exported from endothelial cell tube formation assays from control group exosomes at 6 h (the second experiment pictures, repeat the first time)

Click here for additional data file.

Supplemental Information 53 Raw data exported from endothelial cell tube formation assays from p62 knockdown group exosomes at 6 h (the first experiment pictures, repeat the first time)

Click here for additional data file.

Supplemental Information 54 Raw data exported from endothelial cell tube formation assays from p62 knockdown group exosomes at 6 h (the third experiment pictures, repeat the first time)

Click here for additional data file.

Supplemental Information 55 Raw data exported from endothelial cell tube formation assays from p62 knockdown group exosomes at 6 h (the second experiment pictures, repeat the first time)

Click here for additional data file.

Supplemental Information 56 Raw data exported from endothelial cell tube formation assays from p62 knockdown group exosomes at 6 h (the second experiment pictures, repeat the first time)

Click here for additional data file.

Supplemental Information 57 Raw data exported from endothelial cell tube formation assays from p62 knockdown group exosomes at 6 h (the third experiment pictures, repeat the first time)

Click here for additional data file.

Supplemental Information 58 Raw data exported from endothelial cell tube formation assays from p62 knockdown group exosomes at 6 h (the third experiment pictures, repeat the third time)

Click here for additional data file.

Supplemental Information 59 Raw data exported from endothelial cell tube formation assays from p62 knockdown group exosomes at 6 h (the third experiment pictures, repeat the fourth time)

Click here for additional data file.

Supplemental Information 60 Raw data exported from endothelial cell tube formation assays from HUVEC exosomes at 6 h (the second experiment data, repeat the second time)

Click here for additional data file.

Supplemental Information 61 Raw data exported from endothelial cell tube formation assays from p62 knockdown group exosomes at 6 h (the fourth experiment pictures, repeat the second time)

Click here for additional data file.

Supplemental Information 62 Raw data exported from endothelial cell tube formation assays from p62 knockdown group exosomes at 6 h (the fourth experiment pictures, repeat the first time)

Click here for additional data file.

Supplemental Information 63 Raw data exported from endothelial cell tube formation assays from U-937 exosomes at 6 h (the first experiment pictures, repeat the second time)

Click here for additional data file.

Supplemental Information 64 Raw data exported from endothelial cell tube formation assays from U-937 exosomes at 6 h (the first experiment pictures, repeat the second time)

Click here for additional data file.

Supplemental Information 65 Raw data exported from endothelial cell tube formation assays from p62 knockdown group exosomes at 6 h (the fourth experiment pictures, repeat the third time)

Click here for additional data file.

Supplemental Information 66 Raw data exported from endothelial cell tube formation assays from U-937 exosomes at 6 h (the first experiment pictures, repeat the first time)

Click here for additional data file.

Supplemental Information 67 Raw data exported from endothelial cell tube formation assays from U-937 exosomes at 6 h (the second experiment pictures, repeat the second time)

Click here for additional data file.

Supplemental Information 68 Raw data exported from endothelial cell tube formation assays from U-937 exosomes at 6 h (the second experiment pictures, repeat the first time)

Click here for additional data file.

Supplemental Information 69 Raw data exported from endothelial cell tube formation assays from U-937 exosomes at 6 h (the second experiment pictures, repeat the first time)

Click here for additional data file.

Supplemental Information 70 Raw data exported from endothelial cell tube formation assays from U-937 exosomes at 6 h (the secnd experiment pictures, repeat the second time)

Click here for additional data file.

Supplemental Information 71 Raw data exported from endothelial cell tube formation assays from U-937 exosomes at 6 h (the third experiment pictures, repeat the second time)

Click here for additional data file.

Supplemental Information 72 Raw data exported from endothelial cell tube formation assays in HUVEC at 4 h using 20x microscopic (the fourth experiment pictures, repeat the second time)

Click here for additional data file.

Supplemental Information 73 Raw data exported from endothelial cell tube formation assays from control group exosomes at 4 h using 20x microscopic

Click here for additional data file.

Supplemental Information 74 Raw data exported from endothelial cell tube formation assays from control group exosomes at 4 h using 10x microscopic

Click here for additional data file.

Supplemental Information 75 Raw data exported from endothelial cell tube formation assays from p62 knockdown group exosomes at 4h using 10x microscopic

Click here for additional data file.

Supplemental Information 76 Raw data exported from endothelial cell tube formation assays from p62 knockdown group exosomes at 4h using 40x microscopic

Click here for additional data file.

Supplemental Information 77 Raw data exported from endothelial cell tube formation assays from p62 knockdown group exosomes at 4h using 20x microscopic

Click here for additional data file.

Supplemental Information 78 Raw data exported from endothelial cell tube formation assays from control group exosomes at 4 h using 40x microscopic

Click here for additional data file.

Supplemental Information 79 Raw data exported from endothelial cell tube formation assays from U-937 group exosomes at 4 h using 20x microscopic

Click here for additional data file.

Supplemental Information 80 Raw data exported from endothelial cell tube formation assays in HUVEC at 3 h using 10x microscopic

Click here for additional data file.

Supplemental Information 81 Raw data exported from endothelial cell tube formation assays in HUVEC at 3 h using 40x microscopic

Click here for additional data file.

Supplemental Information 82 Raw data exported from endothelial cell tube formation assays from U-937 group exosomes at 4 h using 40x microscopic

Click here for additional data file.

Supplemental Information 83 Raw data exported from endothelial cell tube formation assays in HUVEC at 4 h using 20x microscopic

Click here for additional data file.

Supplemental Information 84 Raw data exported from endothelial cell tube formation assays in HUVEC at 3 h using 20x microscopic

Click here for additional data file.

Supplemental Information 85 Raw data exported from endothelial cell tube formation assays from control group exosomes at 4 h using 10x microscopic

Click here for additional data file.

Supplemental Information 86 Raw data exported from endothelial cell tube formation assays from control group exosomes at 4 h using 20x microscopic

Click here for additional data file.

Supplemental Information 87 Raw data exported from endothelial cell tube formation assays from control group exosomes at 4 h using 40x microscopic

Click here for additional data file.

Supplemental Information 88 Raw data exported from endothelial cell tube formation assays from p62 knockdown group exosomes at 4h using 40x microscopic

Click here for additional data file.

Supplemental Information 89 Raw data exported from endothelial cell tube formation assays from p62 knockdown group exosomes at 4h using 10x microscopic

Click here for additional data file.

Supplemental Information 90 Raw data exported from endothelial cell tube formation assays from p62 knockdown group exosomes at 4h using 20x microscopic

Click here for additional data file.

Supplemental Information 91 Raw data exported from endothelial cell tube formation assays from U-937 group exosomes at 4 h using 10x microscopic

Click here for additional data file.

Supplemental Information 92 Raw data exported from endothelial cell tube formation assays in HUVEC at 4 h using 10x microscopic

Click here for additional data file.

Supplemental Information 93 Raw data exported from endothelial cell tube formation assays from U-937 group exosomes at 4 h using 20x microscopic

Click here for additional data file.

Supplemental Information 94 Raw data exported from endothelial cell tube formation assays in HUVEC at 4 h (repeat the first time)

Click here for additional data file.

Supplemental Information 95 Raw data exported from endothelial cell tube formation assays in HUVEC at 4 h using 40x microscopic (repeat the first time)

Click here for additional data file.

Supplemental Information 96 Raw data exported from endothelial cell tube formation assay in HUVEC at 4 h using 20x microscopic (repeat the first time)

Click here for additional data file.

Supplemental Information 97 Raw data exported from endothelial cell tube formation assays from control group exosomes at 4 h using 10x microscopic

Click here for additional data file.

Supplemental Information 98 Raw data exported from endothelial cell tube formation assays from control group exosomes at 4 h using 20x microscopic

Click here for additional data file.

Supplemental Information 99 Raw data exported from endothelial cell tube formation assays from p62 knockdown group exosomes at 4h using 10x microscopic

Click here for additional data file.

Supplemental Information 100 Raw data exported from endothelial cell tube formation assays from control group exosomes at 4 h using 40x microscopic

Click here for additional data file.

Supplemental Information 101 Raw data exported from endothelial cell tube formation assays from p62 knockdown group exosomes at 4h using 40x microscopic

Click here for additional data file.

Supplemental Information 102 Raw data exported from endothelial cell tube formation assays from p62 knockdown group exosomes at 4h using 20x microscopic

Click here for additional data file.

Supplemental Information 103 Raw data exported from endothelial cell tube formation assays from U-937 group exosomes at 4 h using 10x microscopic

Click here for additional data file.

Supplemental Information 104 Raw data exported from microarray analysis

Click here for additional data file.

Supplemental Information 105 Raw data exported from microarray analysis

Click here for additional data file.

Supplemental Information 106 Raw data exported from endothelial cell tube formation assays in HUVEC at 4 h using 40x microscopic

Click here for additional data file.

Supplemental Information 107 Raw data exported from endothelial cell tube formation assays from U-937 group exosomes at 4 h using 20x microscopic

Click here for additional data file.

Supplemental Information 108 Raw data exported from endothelial cell tube formation assays in HUVEC at 4 h using 10x microscopic

Click here for additional data file.

Supplemental Information 109 Raw data exported from endothelial cell tube formation assays from U-937 group exosomes at 4 h using 40x microscopic

Click here for additional data file.

Supplemental Information 110 Miame Checklist

Click here for additional data file.

Additional Information and Declarations

Competing Interests

Author Contributions

Microarray Data Deposition

Data Availability

The authors declare there are no competing interests.

Chuan Li conceived and designed the experiments, performed the experiments, prepared figures and/or tables, authored or reviewed drafts of the article, and approved the final draft.

Xinyi Long conceived and designed the experiments, performed the experiments, prepared figures and/or tables, and approved the final draft.

Peiqi Liang analyzed the data, prepared figures and/or tables, and approved the final draft.

Zhuogang Liu conceived and designed the experiments, analyzed the data, prepared figures and/or tables, authored or reviewed drafts of the article, and approved the final draft.

Chen Wang analyzed the data, prepared figures and/or tables, and approved the final draft.

Rong Hu conceived and designed the experiments, prepared figures and/or tables, authored or reviewed drafts of the article, and approved the final draft.

The following information was supplied regarding the deposition of microarray data:

The data is available at GEO: GSE192628.

https://www.ncbi.nlm.nih.gov/geo/query/acc.cgi?acc=GSE192628.

The following information was supplied regarding data availability:

The data are available in the Supplementary Files.

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
