# Peer review of "Analysis of microRNA expression profiles in exosomes derived from acute myeloid leukemia by p62 knockdown and effect on angiogenesis"

_PeerJ, doi:10.7717/peerj.13498_

## Round 0.1 · original submission · Major Revisions

Please address the concerns of both reviewers and amend the manuscript accordingly.

·

Basic reporting

Paper titled “Analysis of microRNA expression profiles in exosomes derived from acute myeloid leukemia by p62 knockdown and effect on angiogenesis” by Li et al is a research paper that investigates how the microRNA profiles change in exosomes upon p92 knockdown and the effect of these exosomes in angiogenesis.

I suggest that the paper needs to be revised to be considered for publication.

The positive aspect of the paper is that it contains microarray analysis that surveys the expression profiles of thousands of exosome microRNAs and uses pathway databases to predict networks and miRNA function. In this aspect the paper can be valuable to literature.


1) The language of the paper is not clear and often times confusing to the reader. There are also grammar errors especially in results and methods section. I suggest the authors to have a colleague who is proficient in English and familiar with the subject matter to review their manuscript or contact a professional editing service. For example, I think lines 41 and 42 sound contradicting. In line 86 using “triplicate” is redundant.

Experimental design

8) In methods section, how proteins were extracted from U937 cells for Western Blot (WB) analysis needs to be described. The specific identity of the antibodies used in WB analysis and where they were obtained from needs to me mentioned. In the text, there are no references for the Trizol method, kits, softwares and databases that were used in this study. These references should be added. In section 7 of methods, it is not clear how the normalization was done, please clarify.
12) In figure legends please include how many replicates were used to achieve the error bars.
13) Include dates in your references so that the readers can easily see how recent is the knowledge that you are referring to.
14) In 5th section of the results, please explain, what is targetscan7.1 and mirdbV5 are and how do these databases differ from each other.
15) Introduction requires more detail, with appropriate references, on the importance of P62. The specific aims and research questions should be stated in the introduction.
16) I would like authors to give more information on U937 cells and why they chose these cells for this study.

Validity of the findings

2) BCA does not seem to be the appropriate method for exosome quantification. I suggest repeating these experiments using more appropriate technique(s). One suggestion for these techniques might be https://www.ncbi.nlm.nih.gov/pmc/articles/PMC5039048/
3) As part of figure 4, some of the miRNAs that were found to be upregulated or downregulated should be tested with experimentation such as RT-qPCR and WB and these results should be included in the paper.
4) Figure 6 needs quantitation of angiogenesis. The scale bar in figure 6 is unreadable and it looks like the scale bars might not be consistent amongst the images. Please address this issue. I don’t agree with lines 204-205 and lines 275-277 in the discussion. I think in 6B, HUVECs+Exo(U937), HUVECs+Exo(p62-con) and HUVECs+Exo(p62-) all look very similar to each other.
5) The experimental evidence is not shown for the lines 257-266 of discussion. Please label figures 4 and 5 as suggested below for clarification.
6) I suggest the section 4 of results to be re-written as it was hard for me to follow the text. Labels in figure 4 are not seen at all and needs to be fixed.
7) Figure 5 needs better resolution. C and d are unreadable and there are no labels in e and f which prevents the reader from understanding the elements of the network and their relation to each other.
9) Figure 1C, needs quantitation of WB for p62 expression. Figures 1 d,e and f are not explained in the text, please include them in the text. For example, what does the cell percentages mean in 1f and what does the differences in these percentages amongst the three different samples mean?
10) Figure 2B also needs quantitation of WB. It is not clear to me as to what is the difference between concertation of exosomes and quantities of exosomes, please elaborate.
11) In figure 3, please include a legend for the heat map so that the readers can understand where miRNA-3064-3p and miR-339-5p are.

Reviewer 2 ·

Basic reporting

Authors have miRNA in exosomes derived from AML cells after p62 knockdown. The manuscript is very well organized and easy to understand. I have few minor comments.
1. Is it possible to include a little more literature survey on the miRNAs? The biological role of miRNA has been well described. But some information about current pharmaceutical research targeting miRNA (if at all there is anything) would be helpful.
2. The figure legends for figures 4,5,6 are too small. Is it possible to increase the size so that it can be easily read.
3. In line 214, 'is' is missing between "p62" and "involved"
If authors can take care of these minor comments, I recommend accepting it for publication.

Experimental design

To the point.

Validity of the findings

properly described and interesting.

---

## Round 0.2 · Minor Revisions

Please note that although both reviewers are mostly satisfied by the revision, reviewer #1 indicated that labels in figures 4, 5 and 6 cannot be seen. Please fix this issue.

·

Basic reporting

Language of the paper has improved.

Experimental design

The usage of BCA has been justified by the authors in their rebuttal.

Validity of the findings

The findings are valid, prior concerns were addressed by the authors.

Additional comments

I thank the authors for the changes thy made in their manuscript and their responses in the rebuttal. Overall, I think the paper has significantly improved and t I can still not see the labels in figures 4, 5 and 6 even when I zoom in. I think the representation of these figures still have room for improvement.

Reviewer 2 ·

Basic reporting

Up to the mark

Experimental design

Up to the mark

Validity of the findings

Up to the mark

Additional comments

The authors took care of my previous comments.

---

## Round 0.3 · accepted · Accept

All remaining issues were addressed and the revised manuscript is acceptable now.